# Trustworthiness and a Zero Leakage OTMP-P2L Scheme Based on NP Problems for Edge Security Access [note 1]

**DOI:** 10.3390/s20082231

**Published:** 2020-04-15

**Authors:** Daoqi Han, Xiaofeng Du, Yueming Lu

**Affiliations:** 1Key Laboratory of Trustworthy Distributed Computing and Service, Ministry of Education, Beijing University of Posts and Telecommunications (BUPT), Beijing 100876, China; handq@bupt.edu.cn (D.H.); duxiaofeng@bupt.edu.cn (X.D.); 2School of Information and Communication Engineering, Beijing University of Posts and Telecommunications (BUPT), Beijing 100876, China; 3College of Cyberspace Security, Beijing University of Posts and Telecommunications (BUPT), Beijing 100876, China

**Keywords:** chained argument, consensus mechanism, edge security access, multitask proofs, zero knowledge proof

## Abstract

Resource constraints have prevented comprehensive cryptography and multifactor authentication in numerous Internet of Things (IoT) connectivity scenarios. Existing IoT systems generally adopt lightweight security protocols that lead to compromise and privacy leakage. Edge computing enables better access control and privacy protection, furthermore, blockchain architecture has achieved a trusted store of value by open-source and distributed consensus mechanisms. To embrace these new paradigms, we propose a scheme that employs one-time association multitasking proofs for peer to local authentication (OTMP-P2L). The scheme chooses relevant nondeterministic polynomial (NP) problem tasks, and manages localized trust and anonymity by using smart devices such as phones and pads, thereby enabling IoT devices to autonomously perform consensus validation with an enhanced message authentication code. This nested code is a one-time zero-knowledge proof that comprises multiple logic verification arguments. To increase diversity and reduce the workload of each one, these arguments are chained by a method that establishes some of the inputs of the following task from the output of previous tasks. We implemented a smart lock system and confirmed that the scheme outperforms IoT authentication methods. The result demonstrates superior flexibility through dynamic difficulty strategies and succinct non-interactive peer-to-peer (P2P) verification.

## 1. Introduction

Current IoT systems frequently encounter cyber-attacks, including distributed denial of service (DDoS) attacks, exploits, and viruses. These attacks have caused severe infrastructure damage and information leaks. Cyber-attacks usually target small IoT devices like shared smart cars, printers, webcams, residential gateways, medical devices, and smart grids. These weak devices, whose security has been neglected, are now working closely with humans and infrastructure systems, due to extensive connections, and are also exposed to increasingly sophisticated cyber-attack tools.

The widespread deployment of IoT devices has overwhelmed cloud computing centers. This trend drives the rise of edge computing frameworks. The new framework has extended cloud computing by moving computing and data services to the edge of clouds [1] and provides intelligent services, low-latency storage, and data processing for resource-limited Internet of Things (IoT) nodes. Many novel optimization solutions balance edge and cloud computing for optimal response times and cost [2,3]. Traditionally, IoT devices sent sensed data to and/or acquired data and services directly from the cloud, which was time-consuming. Certain smart applications, especially in the health-care or emergency sectors could not tolerate such delays. With the introduction of fog and edge computing, data analysis and service acquisition are processed at a faster pace [4]. Moreover, breakthroughs in chip technology have significantly enhanced the functionality of smart devices. With these technologies, a local, centralized security center can be created for trust and security management in the IoT. The powerful local center will be able to enhance the privacy, authentication, and authorization capabilities of the IoT, and create demilitarized zones for trillions of devices.

Blockchain technology has been widely validated for solving problems such as distributed consensuses, trust authentication, privacy protection, cyberattacks, and cracking. Through a full-featured peer-to-peer client software, this technology can establish a decentralized trust mechanism to deliver values reliably, without involving a centralized third-party system. With readily generalizable technical frameworks, blockchain technology has become an essential strategic technology and has rapidly been implemented in applications like the IoT, e-commerce, e-government, digital medical systems, and power services. In their report on the top-ten strategic technology trends for 2017 [5], Gartner pointed out that the distributed ledger could be a revolutionary technology that represents a trillion-dollar opportunity in the supply chain. In 2018, zero-knowledge proofs were considered [6] the latest breakthrough in blockchain technology.

In the edge security center, using such consensus technology can solve existing IoT device authentication issues, and protect privacy using zero-knowledge proofs. Therefore, we design one-time association multitask proofs for peer to local authentication (OTMP-P2L) (shown in Figure 1), which employs a peer-to-peer (P2P) consensus verification based on the same cryptography techniques. Furthermore, the scheme chooses various nondeterministic polynomial (NP) problems for performing verification tasks, improves diversity, and reduces the computational complexity of verification. Embedded devices receive the messages by a sensor and verify that offline with less computing costs. On the contrary, the powerful smart devices produce one-time electronic tokens as verification elements in a local reservation batch and generate quick response (QR) codes for flexible transfer and sharing. Multifactor authentication and anonymization strengthen the scheme’s robustness against network attacks and physical hijacks, thereby preventing network intrusions, service disruptions, and other risks.

To the best of our knowledge, we are the first to chain zero-knowledge proofs to enhance the message authentication code. The main contributions of this paper can be summarized as follows:We conduct the study on zero-knowledge proof algorithms and propose a feasible multi algorithms chaining scheme. Since the previous message authentication code needs manual interaction, the content is simple, and the algorithm is single. In the open environment of the IoT, it is necessary to enrich the verification content, increase the complexity of verification codes and the difficulty of production, simplify the verification process, and automate. Thus, we design the enhanced message authentication code, which can limit the time, place, identity and collection members by multifactor zero-knowledge non-interactive verification. We choose a variety of chaining NP problems to achieve verification. The new scheme can exponentially increase the difficulty of crack, thereby reducing the workload.We implement an anonymously distributed authorization mechanism that allows users to manage addresses dynamically and choose one autonomously. The centralized authorization has problems such as forgery, single point of failure, user identity, and other data leakage. After removing the user data control right, the authorization center only completes the negotiation and sharing of the common reference string, and the center provides the reservation token service according to the access relationship agreed by the service provider. The provider achieves non-interactive point-to-point verification. So the anonymous mechanism avoids data leakage.We make a smart lock prototype system and deploy on MUC and PC equipment to verify the performance of six test cases. The experimental results show that the amount of data required for the proposed scheme is not substantial, and that the resource usage of the verification calculations is within control. The system can be deployed and run stably in existing smart devices and resource-constrained IoT nodes.

The remainder of this paper is organized as follows: The second section introduces the work related to IoT authentication. In Section 3, we present the system architecture. In Section 4 and Section 5, we describe the details of algorithms and models. The experimental results and a scheme comparison are described in Section 6. We make our conclusions in Section 7.

## 2. Related Work

Shi et al. [1] defined edge computing as the enabling technologies allowing computation to be performed at the edge of the network, as well as discussing typical scenarios for its use, proving its necessity, and introducing possible challenges and opportunities. Lin et al. combined edge computing to reduce delay and bandwidth, with accurate services for user behavior on social networks, proposed innovative vehicle social edge computing (VSEC) [3], and provided theoretical and empirical examples for the applications. They further classified four typical user services, analyzed the detailed characteristics of these services, and calculated an optimal resource allocation strategy using the model [7]. Similarly, security services can also be deployed in edge networks to reduce delay and bandwidth. Furthermore, enhancement of local centers will be more resistant to intrusion than microcontroller unit (MCU) controllers and sensors.

Traditional password authentication methods were first used extensively in human-computer interactions. In order to improve security, these methods adopted multiple factors, such as SMS, personalized information convention and verification, certificates and public keys, and environmental and biometrical characteristics. In contrast, the decentralized blockchain system [8] adopts a simple method that publishes a one-time anonymous signature and succinct workload verification to realize trusted authentication between any parties. Although these systems have different architectures, they both use one-way NP tasks, which are difficult for computers to solve but can be easily verified, to build efficient defense systems.

End-to-end identity authentication of devices in the IoT is challenging, as these devices are exposed to physical and internet environments full of potential attackers. A setting based on physical proximity and an administrator’s personal identification number (PIN) makes it convenient to bind or unbind devices and establish trust relationships but also brings risks, such as middleman attacks and hijacking. Many IoT devices frequently interact with other systems; however, these are merely dynamic automatic and semi-automatic registration mechanisms.

Due to the limited capacity of previous devices, the implemented authentications were fragile. Malware has targeted the IoT, attempting to spread to all devices held by the victim when connected, to further infiltrate to the target device. In addition, malware can lurk in the terminal for long-term analysis and control of IoT devices, such as surveillance cameras. Such malware has become a severe threat to IoT devices. Aiming at these IoT issues, Zhang et al. [9] analyzed the current status of, and research opportunities in, IoT security, suggesting that the differences between IoT and traditional security issues were the heterogeneity and complexity of the former.

Kim et al. [10] proposed that identity and authorization are important components of basic security. Luk et al. identified seven compulsory attributes of identity authentication and analyzed the performance and disadvantages of multiple broadcast authentication protocols [11]. The specific implementation schemes are as follows: Yao et al. described the pros and cons of three certificate authentication schemes based on public key and identity, as well as authentication based on a symmetric key and one-time signature, and proposed a fast one-way accumulator for implementing a lightweight multicast authentication mechanism [12].

All of these schemes adopted lightweight authentication and chose to run complex tasks on the server-side to avoid issues with the device’s power consumption and resource occupation. They realized rudimentary access control, but did not consider problems including DDoS attacks, middleman attacks, and physical threats. Utilizing edge computing theory, the researchers in [13,14,15,16] analyzed typical IoT attack and defense measures and proposed establishing a localized security protection system.

Cryptography technology is also constantly making breakthroughs. The naive consensus mechanism of proof of work (PoW) allows for trusted transactions in open and insecure networks. Courtois et al. [17] suggested using an optimized SHA256 algorithm in bitcoin mining, which could improve its performance by approximately 90%. Koblitz et al. [18] proposed the use of ECC in cryptography; in contrast to traditional RSA algorithms, ECC encryption is more resistant to attacks, and has lower hardware requirements and a faster encryption speed.

With the maturity of blockchain-related technologies, a large number of combination solutions have emerged to solve the problems caused by the centralized structure of the original scenarios. The WLAN mesh network has a distributed network structure that is multi-hop, self-organizing, and self-healing, Xin et al. [19] considered all authentication records in the mesh network as a public ledger to effectively monitor malicious attacks. Zhao et al. [20] proposed a blockchain-based risk assessment architecture. The system is based on an improved DPOS consensus algorithm that can be distributed fault-tolerant and tamper-resistant. The original data content is encrypted and stored in the database, reducing the pressure on the ledger storage. Li et al. [21] proposed an anti-quantum proxy blind signature scheme based on lattice cryptography, which provides user anonymity and untraceability in distributed applications of blockchain-enabled IoT, can resist quantum attacks. It is more common to access sensing data and make decisions through IoT smart devices [22], so guaranteeing user privacy and maintaining the integrity of collected data becomes more important. Wang et al. [23] propose a balanced scheme attempts to preserve a balance in user privacy, data integrity and the computational cost. Tseng et al. [24] proposed a perspective architecture with the hierarchical design, discussing the challenges in the integration of blockchain and IoT, pointing out that it is nontrivial to integrate blockchains and IoT systems.

Zero-knowledge proofs, which were first proposed and conceptualized by Goldwasser et al. [25] in 1989, are defined as proofs that do not convey any knowledge other than the correctness of the proposition. These are full of research results [26,27,28,29] in terms of performance and privacy protection. The proposed OTMP-P2L scheme chooses PoW, hash-based message authentication code (HMAC), one-time signature, one-way accumulator, homomorphic encryption, and zero-knowledge succinct non-interactive arguments of knowledge algorithms (zk-SNARK) to generate encrypted dynamic length proofs that are difficult to solve but are succinct and efficient to verify. By dynamically choosing the type of proofs in the token, we can exponentially expand the difficulty of cracking calculations and achieve flexible access control and privacy protection.

## 3. OTMP-P2L Architecuture

The OTMP-P2L scheme constructs a one-time association multitask proofs (OTMP) set to archive the dynamic combination of proofs as shown in Figure 2. It reduces the computation and energy consumption of IoT devices by the easy-to-verify feature of NP problem, simultaneously establishing a local security center to control the IoT devices within the edge network. Through the early local multifactor authentication process, this guarantees the legitimacy of members, establishes point-to-local center’s secure channels (P2L), and exchanges the shared secret information. The device that, unlike the human who can actively complete authentication at any time, needs to complete authentication in advance and batch apply and store service quantity proofs (SP) tokens by the agent.

### 3.1. Register

Trusted administrators use smartphones or local center devices to install security center software. The software can manage all regional IoT devices and control P2P and local-to-cloud access. Then, a localized defense system is built for IoT devices.

Using Bluetooth, WIFI, and other IoT network protocols, the security center scans devices in the environment, displays them in a list, and adjusts their metadata, such as personalized name and location, according to user requirements to form an overview of the network. Afterward, new administrators and viewers can be added to this network, if their basic information and phone number are configured. All members use the final manually set attribute string to generate an anonymous address as the identity ID.

### 3.2. Local Authentication

The user provides biometric, PIN, SMS, and other data for complete multifactor user authentication, thus achieving security and reliable access. The trusted administrator performs local device identification, pairing, unique ID confirmation, binding, and other device authentication operations locally.

The administrator assigns multiple sets of addresses and key pairs to users and devices to achieve full anonymization within the network. Thus, the security center can establish secure channels, transfer keys, and distribute shared security information.

Once a user or device passes local authentication, only the administrators can view the confidential audit information in the security center software or unbind devices to open access. Otherwise, the devices and users transfer the encrypted data anonymously.

### 3.3. Local Authorization

The administrator configures access between a user and devices, or a device and devices, and agrees to access the shared information. The shared information is synchronized through the secure channel.

By booking services and applying for service quantity proofs (SP) in batches, providers dynamically combine multiple NP tasks to achieve a flexible security level. Through secure channels, an SP can be distributed to the user or device that initiated the application. The shorter the valid period of the SP, the more the number of batches, and the lower the probability of being exploited and cracked.

At present, the set has five types of tasks, time-period proof, legal token proof, workload proof, polynomial proof, and legal member proof. They are implemented via homomorphic encryption, HMAC, ECC, zero-knowledge proof, and one-way accumulator, respectively.

### 3.4. P2P Verification

The provider can flexibly choose different modes of sound, light, or electrical signals and provide the current valid SP and request proofs (RP) through unsecured networks, and the verifier can dynamically execute the verification tasks according to certification requirements. After verification, the verifier performs the service and returns the results.

## 4. OTMP Methodology

The scheme utilizes the shared secret information CRS (common reference string) to produce volume service proofs (SP) in advance, and generate an anonymous signature proof (RP) on each request. In this way, IoT devices only need one-way authentication of encrypted proofs for most of the time, do not need to access the registration and identity authentication information, and do not need to apply for the service proofs frequently. The NP problems that would be cracked in the exponential time are difficult to solve, but easy to verify. This unidirectionality guarantees the security of the token. Multiple NP tasks use different pre-made shared information CRS and different verification methods to improve the security level of the system gradually. The relevant notations in the OTMP-P2L scheme are listed in Table 1.

### 4.1. Methodology

#### 4.1.1. Zero Knowledge Proof

For the interactive proof system (Alice,Bob), Alice is the provider, Bob is the verifier, and Alice is at the definite probability:(1)$$1-\frac{1}{n^{k}}.$$

Then Alice can provide sufficient evidence to prove to Bob the correctness of event *L*, where *n* is the length of the input and *k* is the known amount of quantitative knowledge of Bob. Constructing multiple kinds of evidence makes the forgery more difficult, and the probability of correctness tends to 1.

Zcash [30] uses zk-SNARK to prove that the conditions for valid trading are met without revealing any important information about the address or value involved. It hides the entered address, the output address and the amount, but can guarantee:The addition of input values is the output of every hidden transaction.The sender proves that they have a private key for each entry and therefore has the power to consume.The cost private key of the input item is linked to the signature of the entire transaction in an encrypted way, and it is difficult for anyone can modify the transaction without knowing these private keys.

The core technology is the homomorphism algorithm, and the function E(x) that satisfies the following three conditions is called additive homomorphism.

For most *x*, it is usually difficult to solve *x* at a given E(x).Different inputs will get different outputs, so if x≠y, then E(x)≠E(y).If someone knows E(x) and E(y), E(x) and E(y) can be used to calculate E(x+y).

The three encrypted numbers satisfy the addition formula:(2)E(x)+E(y)=E(x+y).

E(x) and E(y) are cryptographic evidence that proves that someone possesses knowledge x,y and the addition algorithm respectively. By combining the characteristics of additive homomorphism and multiplicative homomorphism, the zero-knowledge proof can be generalized to the polynomial calculation.

Hypothetically, Alice knows a polynomial *P* of the highest *d* times, and Bob wants to know E(P(s)) corresponding to a specific *s*.
(3)$$P\left(x\right)=a_{0}+a_{1}x+a_{2}x^{2}+a_{3}x^{3}+\ldots +a_{d}x^{d}.$$

In the process of verification, Alice only knows *P*, does not know *s*, Bob only knows *s*, does not know *P*, and realizes Bob to obtain E(P(s)) by:For each exponential of *s*, Bob calculates E(1),E(s),…,
$E(s^{d})$ and sends them to Alice.Alice knows all the coefficients of the polynomial and can calculate E(P(s)) using the homomorphic property and send it back to Bob.

The additive homomorphism Formula (Equation 2) implements additive hiding, allowing Bob to check the value of x+y without knowing x and y, respectively. Similarly, the polynomial homomorphism Formula (Equation 3) hides polynomial P and checks the value of P(x+y) without exposing P(x) and P(y).

In order to convert arbitrary calculations into polynomial proofs, quadratic arithmetic programs (QAP) first convert Boolean circuit calculations into polynomial calculations, for example: $S_{1}\;=\;C_{1}\;+\;C_{2},S_{2}=C_{3}*C_{4},S_{3}=S_{1}*S_{2}$; the complex calculation is simplified to the gate expression by adding intermediate variables. The new gate circuit is equivalent to the original calculation. Then, through R1CS (rank-1 constraint system), each new gate circuit is converted into a corresponding vector.

The original vector of the circuit is: $s=[C_{1},C_{2},C_{3},C_{4},$
$S_{1},S_{2},S_{3}]$. For $S_{1},S_{2},S_{3}$, we define three vector groups $(a_{1},b_{1},$
$c1),\left(a_{2},b_{2}c_{2}\right),\left(a_{3},b_{3},c_{3}\right)$.

Corresponding to $a_{1}=\left[1,0,0,0,0,0,0\right],b_{1}=\left[0,1,0,0,0,0,0\right],c_{1}=\left[0,0,0,0,1,0,0\right]$, so $s\times a_{1}+s\times b_{1}=s\times c_{1}$, mean $S_{1}=C_{1}+C_{2}$.

Similarly, the following two sets of vectors are obtained:


$a_{2}=\left[0,0,1,0,0,0,0\right],b_{2}=\left[0,0,0,1,0,0,0\right],c_{2}=\left[0,0,0,0,0,1,0\right];s\times a_{2}*s\times b_{2}=s\times c_{2};S_{2}=C_{3}*C_{4};$



$a_{3}=\left[0,0,0,0,1,0,0\right],b_{3}=\left[0,0,0,0,0,1,0\right],c_{3}=\left[0,0,0,0,0,0,1\right];s\times a_{3}*s\times b_{3}=s\times c_{3};S_{3}=S_{1}*S_{2}.$


Now the Boolean circuit calculation is converted into a vector calculation. Next, the QAP algorithm converts the general vector calculation into a polynomial calculation: We select any value $x_{1}$ on the finite field *F*, according to a set of polynomials:


$a_{1}=\left[P_{a_{1}}\left(x\right),P_{a_{2}}\left(x\right),P_{a_{3}}\left(x\right),P_{a_{4}}\left(x\right),P_{a_{5}}\left(x\right),P_{a_{6}}\left(x\right),P_{a_{7}}\left(x\right)\right];$



$b_{1}=\left[P_{b_{1}}\left(x\right),P_{b_{2}}\left(x\right),P_{b_{3}}\left(x\right),P_{b_{4}}\left(x\right),P_{b_{5}}\left(x\right),P_{b_{6}}\left(x\right),P_{b_{7}}\left(x\right)\right];$



$c_{1}=\left[P_{c_{1}}\left(x\right),P_{c_{2}}\left(x\right),P_{c_{3}}\left(x\right),P_{c_{4}}\left(x\right),P_{c_{5}}\left(x\right),P_{c_{6}}\left(x\right),P_{c_{7}}\left(x\right)\right].$


When $x=x_{1}$, we make the vector calculation corresponding to $S_{1}=C_{1}+C_{2}$. Similarly, $x_{2},x_{3}$ is taken out from *F*, and satisfies $\left(a_{2},b_{2},c_{2}\right),\left(a_{3},b_{3},c_{3}\right))$, respectively.

Now, QAP represents the original three vector groups in three vector groups $\left(a_{1}\left(x_{1}\right),b_{1}\left(x_{1}\right),c_{1}\left(x_{1}\right)\right),\left(a_{2}\left(x_{2}\right),b_{2}\left(x_{2}\right),c_{2}\left(x_{2}\right)\right)$, and $\left(a_{3}\left(x_{3}\right),b_{3}\left(x_{3}\right),c_{3}\left(x_{3}\right)\right)$ represented by *x*.

Let the polynomial $P\left(x\right)=s\prod a_{1}\left(x\right)*s\prod b_{1}\left(x\right)-s\prod c_{1}\left(x\right)$, as defining $x=x_{1},x_{2},x_{3},P\left(x\right)=0$, by polynomial theorem, there exists H(x), so that P(x)=T(x)∗H(x), where $T\left(x\right)=\left(x-x_{1}\right)\left(x-x_{2}\right)\left(x-x_{3}\right)$. By calculating the polynomial, P(x)=T(x)∗H(x) we can verify the original equation, which is exactly the original circuit.

Generalized to general computing, QAP consists of a set of polynomials and a linear combination of tasks that find multiples of a given polynomial. The QAP problem on the domain *F* with an input length of *n* consists of the following three parts:A set of polynomials over a finite field *F*: $v_{0},v_{1}\ldots vm,w_{0},w_{1}\ldots wm$;The target polynomial *t* on the finite field *F*;The injective function f:{(i,j)∣1≤i≤n,j∈{0,1}}→{1,…,m}.

An input *u* is accepted by the QAP, if and only if the vector group $a=\left(a_{1},a_{2},\ldots ,a_{m}\right),b=\left(b_{1},b_{2},\ldots ,b_{m}\right)$ in the finite field F matches the following conditions:If k=f(i,u[i]), then k=f(i,u[i]); (where u[i] is the i-th bit of *u*);If k=f(i,1−u[i]), then $a_{k},b_{k}=0$;The target polynomial *t* can be divisible by $v_{a}w_{b},v_{a}=v_{0}+a_{1}v_{1}+\ldots +a_{m}v_{m},w_{b}=w_{0}+b_{1}w_{1}+\ldots +b_{m}w_{m}$.

All verifiers must check whether the polynomial *t* can be divisible by $v_{a}w_{b}$.

#### 4.1.2. Proof of Work

The main principle of PoW is to retrieve whether a certain number of bits in front of the hash value are zero. After adding a random number (nonce) to each block of information, the hash value generated by SHA256 encryption can be found to start with a certain number of zeros and then be verified and broadcast in the blockchain to be packaged into the ledger. For each digit, the amount of computation increases exponentially, and the longest chain represents the consensus result that the maximum workload votes.

Bitcoin gives a fast logarithm algorithm for calculating the difficulty value by means of the Taylor series variant. The difficulty value is adjusted every 2016 blocks and is calculated according to the block time of the previous 2015 blocks. The formula is as follows:(4)d=p×n/t,
where *d* is the current block target value of calculation difficulty (difficulty), *p* is the previous block target value of calculation difficulty (previous target), *t* is the time taken for the previous 2015 blocks’ generation, and *n* is the constant 1,209,600, that is, the number of seconds required to generate 2016 blocks, a block every 10 min according to the standard [8].

As it is a deterministic algorithm, all nodes have the same calculation difficulty, and a new block can be generated every 10 min. In order to forge a block, all blocks must be recalculated with the same difficulty, and the attackers must have the same computational power as the whole network to make it possible. With reference to this idea, we can dynamically set the computational difficulty value of consensus on a verification according to the actual security level requirements, and the attacker cannot pass the verification without paying a certain amount of computing power.

#### 4.1.3. ECC Algorithm

Group definition GF(p) based on an elliptic curve over finite fields can calculate Q=kG. Knowing the public key *Q* and the base point *G*, it is difficult to directly calculate the private key *k* with a similar division method, which is the NP problem. However, knowing the private key *k* and the base point *G*, addition and multiplication (the addition of integer multiples *G*) can easily calculate the encryption result, which is the *P* problem, and satisfies the commutative law and the associative law of Abel Group, regardless of the order of the operation. The ECC algorithm is one order of magnitude of a smaller size than RSA and other asymmetric algorithms under the same security level, thus it has great advantages in generating keys, encryption and decryption, space occupation, and transmission.

In the encryption and decryption scenario, the algorithm is described in the following steps:While encrypting with the public key *Q*, find the random number *M* and calculate $c_{1}=G\times M$ and $c_{2}=Q\times M+x$ (*M* is randomly selected, *x* is the number to be encrypted).While decrypting with the private key *k*, the ciphertexts $c_{1}$ and $c_{2}$ are obtained, calculate $c_{2}-c_{1}\times k=Q\times M+x-G\times M\times k=k\times G\times M+x-G\times M\times k=x$, and the plaintext *x* is decrypted.Even though other attackers know $c_{1}$ and *G*, they cannot directly calculate *M* by division. They also cannot directly calculate *x* with $c_{2}$ and *Q*. Random number *M* can make the values of $c_{1}$ and $c_{2}$ different each time.

In the signature scenario, the algorithm is described in the following steps:While signing with the private key *k*, find the random number *M*, the hash value *h* of the message *x*, the coordinate position $G_{x}$ of the *G*, and calculate $c_{1}=G\times M$ and $c_{2}=\left(h+kG_{x}\right)/M$.While verifying the signature with the public key *Q*, the message *x* and the signature $c_{1},c_{2}$ are obtained, use the hash value *h* of the message *x*, and calculate $\left(hG+G_{x}Q\right)/c_{2}=\left(hG+G_{x}kG\right)/c_{2}=\left(\left(h+kG_{x}\right)G\right)/c_{2}=\left(M\left(h+kG_{x}\right)G\right)/\left(h+kG_{x}\right)=G\times M$. If the calculation result is the same as the received signature $c_{1}$, the verification is successful.

### 4.2. Tasks Set

The tasks realize the legal identity and legal token of the requester by the non-interactive authentication without revealing any important information. According to Formula (Equation 1), the more difficult the verification, the more the probability of correctness of event *L* tends to 1.

The security center stipulates relevant pre-made sharing information during the certification phase. Through the encrypted data in the token SP, the device can prove that the owner’s security center produces the token SP, and the user has authorization for the device within a period. Each time the service volume proofs SP is reserved, the following verifiable facts produce the cryptographic token SP:

#### 4.2.1. Proof of the Period

The information that needs to be hidden by the homomorphic hidden function *E* is the SP generation time gt, time interval it, and current service start time bt. The device calculates E(gt+it) by the received E(gt−x),E(it+x). If it equals the received E(bt), and the truth gt+it=bt can be verified, then *x* is a random number.

The ECC encryption algorithm is also an additive homomorphism algorithm. As described in above, calculating $c_{2}-c_{1}\times k$ can eliminate pulice item k×G×M. When decrpyting $E\left(x_{1}\right)+E\left(x_{2}\right)$, $k\times G\times (M_{1}+M_{2})$ is eliminated and the result is $x_{1}+x2$. The function E(x) is just the ECC encryption algorithm.

#### 4.2.2. Proof of the Legal Token

It is necessary to prove that the user has a service volume SP through the reservation process, as then the device can grant the user Alice the right to use the service of Bob. There are two corresponding proofs for the signature string and the encrypted string: (i) The user private key is used to generate the signature of the hash (contents of the behavior string, CBS), and only the user public key can be used to verify the signature. (ii) The device public key issues a token for the hash (CBS), and only the device private key can decrypt the token.

#### 4.2.3. Proof of Work

The security center uses the SHA256 algorithm to calculate the consensus string for the reservation request as part of the token SP.

#### 4.2.4. Proof of zk-SNARK

The security center selects the random point *x* and the random coefficient *k* to pre-produce the shared information as the CRS. *E* is the value of the two sets of encryption functions: $E\left(1\right),E\left(x\right),E\left(x^{2}\right),E\left(x^{3}\right);E\left(k\right),E\left(kx\right),E\left(kx^{2}\right),E\left(kx^{3}\right)$. For the registration and access relationship binding of each device, the corresponding polynomial P(x) is produced.

First step: Alice has P(x) and produces a random offset polynomial R(x), and calculates the value of two sets of encryption functions according to the CRS:


E(P(x)),E(R(x)),E(P(x)+R(x));



E(kP(x)),E(kR(x)),E(kP(x)+kR(x)).


Second step: with Alice’s two sets of polynomials hidden by a homomorphic crypt function to calculate the result numbers, Bob can verify whether it has met the homomorphic additive Formula (Equation 2). If the proof passes, the function returns True, otherwise returns False.

#### 4.2.5. Proof of Legal Members

The Nyberg one-way accumulator is constructed based on a general hash function, which is efficient in simple bit operation. Its absorption formula is as follows:(5)$$H^{Nyb}\left(H^{Nyb}\left(K_{a},y_{1}\right),y_{1}\right)=H^{Nyb}\left(K_{a},y_{1}\right)=A.$$

Using absorptivity of function $H^{Nyb}$, the accumulated value $A=H^{Nyb}\left(K_{a},y_{1}\right)$ can be treated as proof for each accumulation term. Meanwhile, $H^{Nyb}$ can be the verification function. If the accumulation term $y_{1}$ is an element of the accumulated value *A*. Then, after the verification, the result is still *A*. Using this principle, after the user *U* completes the identity authentication, the accumulated value *A* is generated as the pre-made shared information CRS, and the device can quickly verify whether the user *U* is an item of the accumulated value *A* by the function $H^{Nyb}$.

### 4.3. Verification

#### 4.3.1. SP Reservation

The content of the behavior string CBS, which represents the reservation activity, is defined as the following fields: service device Anonymous ID (AID), authorized user or device AID, service time range, authorized function name, reservation remarks, and the timestamp of reservation.

When the user makes their reservation, first, the behavior string is hashed to the hash contents of the behavior string (HCBS), then the HCBS is encrypted into a signature data (only the center public key and the user public key can verify the signature) twice by the center private key and the user private key, finally, the HCBS is encrypted into a token (only the corresponding device private key can decrypt) by the public key of the device.

To increase randomness and crack the difficulty, the system still needs to find eight random numbers based on the user public key, HCBS, and start and end timestamp, and then calculate the consensus hash string of the PoW mechanism HPC (such as the first 24 bits are zero) by high-speed equipment and certain workload difficulty.

The quick response (QR) code picture (QRP) is generated by the result string of the above security processing include the user’s public key, signature data, token, start and end timestamp, four random numbers, difficulty, and consensus hash string HPC. The purpose of retaining the four random numbers is to let the opponent calculate a certain amount of work and verify the consensus string HPC.

#### 4.3.2. SP Authentication

First, the user provides the QR code and triggers the consensus device to scan the QRP to obtain the data. Then we enter the consensus lock verification phase:After receiving the data, the device uses the internally stored elliptic curve parameter and the private key to decrypt the token, so obtains the hash data HCBS. The algorithm guarantees that only the specified device can decrypt and find the HCBS.The device uses the received user public key, the internal center public key, decrypts the signature data twice, and then verifies that it is consistent with the above HCBS signature.The device uses the user public key, HCBS, start and end timestamps, four known random numbers, and also needs to find the remaining four random numbers, and calculates the consensus hash string HPC of the PoW mechanism with the specified difficulty requirement. If it is consistent with the HPC obtained by the scan, it is verified that both the server and the device have paid the corresponding difficulty work, and the authentication is passed.

For the time limit, the device timestamp control mechanism can automatically invalidate the QR code. The token SP submitted by the device is also the same verification process.

#### 4.3.3. RP Generation and Verification

According to the content of each request message, the ECC algorithm and a temporary private key of AID generates an anonymous signature RP. The service side uses AID and the temporary public key of the requester to verify the signature RP.

### 4.4. Challenges

In an open IoT environment, the main challenges are from the heterogeneity, the large scale of objects, and open physical access. We will carefully analyze the response means of various attacks.

The attacker collects data by eavesdropping. The SP token is a one-time evidence of encryption that cannot be solved to obtain the pre-made shared information CRS and forge a new token. Due to the timestamp control mechanism, each request message and signature token RP are different. Moreover, messages are encrypted by a secure ECC algorithm.Man-in-the-middle attack. When masquerading, there is no trust relationship with the security center. When applying for a legal token SP, an attacker cannot pretend to be a requester to send a message. When replaying a message, it is necessary to update the timestamp, so the token RP also needs to be regenerated.DDoS attack. There is no centralized authentication service, or direct point-to-point authentication. When a device is attacked, it does not affect other devices.Physical hijacking. It is impossible to pass the multifactor authentication and release the original binding relationship, so the pre-made shared information CRS cannot be read and decrypted. When a forged CRS is transmitted to the device through a secure channel, it also cannot pass the authentication.Fictional legal user. Due to the lack of trust with the security center, an attacker cannot apply for legal token SP in a batch. The accumulated value A is generated by using the Nyberg one-way accumulator as the shared information CRS generated by the identity authentication, and the accumulated function can quickly verify whether the opponent is an accumulated item of the accumulated value A.Privacy query. Only the smart device that becomes the security center can query corresponding devices according to the anonymous ID, can check resources, and can audit logs. All the devices use the AID in the verification phase.

We also found some issue of the schema as follow:The multiple proofs will increase program complexity and length of verification strings.Important is also the tradeoff between chain more algorithms and reduce the workload of each one, we should define the security level of each combination before its employment in practical.The static CRS are the keys to hiding knowledge in these algorithms, it is also a potential risk.The batch reservation method places restrictions on the service capabilities of edge devices, and the pre-made tokens cannot bind and protect dynamic real-time data.

## 5. P2L Methodology

Devices and users shared the secret information (CRS) through the local security center. Hence, trust relationships and secure channels can be established between the center and each node (P2L). The center transfers the CRS to activate devices and establishes a mechanism capable of offline consensus. Figure 3 shows an example of a user using a QR code token (SP) to unlock a smart lock [31].

The figure depicts the two-stage process. The administrator first registers the organization’s room number, the lock number, and the phone number of the person who can make the reservation. He then distributes the keys and algorithm parameters, setting them to the lock and the security center. Only when the reservation behavior contains all these critical fields can the lock calculate the validity of the verification input string.

When a user makes a reservation, the security center produces a token based on the organization’s key, the lock’s key, and the behavior string. Using the behavior string, the offline lock can then directly calculate the relevant data via the internally stored key to validate the legitimacy of the token (which cannot be forged without knowing the preset secret). Lastly, the PoW mechanism is adopted to reach a consensus on the secret related to the reservation.

Similarly, for other application scenarios, smartphones and multifactor methods can be used to authenticate the identities of users and devices and to bind the device access relationship. Tokens are reserved for production through the security center with powerful computing. Resource-constrained IoT devices can submit tokens for P2P verification succinctly, thereby saving computing power and network access power.

The local security center is composed of three modules as shown in Figure 4 and are described as follows:

Data management: This module configures the system and resource parameters. In a scenario that requires higher levels of security, it can be configured to exchange audit logs and blacklists with the consensus devices periodically.

Key management: This module manages the keys of security centers, users, administrators, and devices. Then, it should exchange data about activated resources with the data and reservation management modules. This module also performs daily backend management for the security center, including batch activation of devices. Then, the module can provide services to the reservation management module and generate token strings based on the reservation behavior string.

Reservation management: This module adds new device types. For new types of devices, this module can accept data created during batch enabling of devices (e.g., room location, administrator activation data, device number, and authorized user and device information) and then call the key management interface to generate a batch of QR code tokens (SPs) according to the reservation behavior string.

However, the local center store data based on the MySQL, it should upgrade to blockchain-based to achieve open and tamper-resistant store. The single-node reservation service needs to enhance by using P2P communication. Furthermore, it is necessary to achieve a gateway that can make more interactive and protect data between the local and cloud.

## 6. Evaluation

### 6.1. Implementation

We made a prototype of the OTMP-P2L scheme using the smart lock scenario.

The software architecture. The code structure consists of four basic parts: an algorithm service module that implements the underlying security algorithm and encapsulates the application programming interface (API). Data management module. Key management module. Evidence string production, conversion, verification module. Based on these basic functions, the reservation management system provides web services on the server side, and the device control system controls the input, verification, and actions on the ARM control side. All of the above programs are implemented by the go language coding. Based on the Vue front-end framework to complete human-computer interaction, we provide two types of operation interfaces that include the WeChat Subscription and the mobile phone page. Users can use the mobile phone to reserve the room at any time to obtain the verification QR code, submit the verification QR code to trigger the smart lock offline verification, and unlock.

The hardware architecture. The QR code scanner, as shown in Figure 5(left), obtains an image, decodes it, converts it into a string, and outputs it to the program currently reading the keyboard through simulating the keyboard device. The control panel, as shown in Figure 5(middle), drives the DC motor to rotate the switch. It uses an auto-rechargeable battery for a stable power supply. It can be connected to the screen for easy maintenance. So the system can be initialized via a USB flash disk at first. The overall device circuits are shown in Figure 5(right). This scheme simplifies the structure of the smart lock, and only requires collections such as images, Bluetooth, and sound, and can extract the verification string and verify the consensus algorithm in close range. With the centralized management of mobile phones, it undertakes the functions of local key storage and multifactor authentication.

### 6.2. Complexity and Experiments

When there is a context association between tasks, the output of the previous verification task is required to execute the following task. For the three NP tasks connected in a series in this way, the complexity of the solution is calculated by multiplying the exponential complexity of each assignment:(6)$$O(n^{x\times y\times z}).$$

Assuming that three NP verification tasks are randomly selected, the complexity of brute force is $O(n^{x})$, $O(n^{y})$, and $O(n^{z})$. When the three tasks are not related, the complexity is $O(n^{\left(x+y+z\right)})$. However, when these three tasks are associated with a context, the complexity increases in a nested exponential manner. For example, the OTMP scheme must first complete the proof of the period task and get the bt variable. The bt variable is a part of the signed string of the second proof of the legal token task to complete the signature. After the second task executes, the behavior hash string can be decrypted, together with the bt variable, as the consensus source for the third proof of work task. The advantage of this association is that we can reduce the difficulty of each task without reducing the overall security. Furthermore, when the set of such zero-knowledge verification tasks is sufficient and efficient enough, it can be modeled really like a maze, and the result of each step dynamically determines the next task.

In the period of batch production tokens, the calculation of proof and the distribution of tokens to legitimate requesters are completed at the local security center. Proof generation for the given period, the complexity of which depends on the homomorphic addition process of ECC, can be completed in milliseconds. Proof generation for legitimate tokens, whose complexity depends on the encryption and signature algorithm process of ECC, can also be completed in milliseconds.

The PoW proof generation, whose complexity depends on the exponential calculation of the SHA256 algorithm and the digit capacity of the hash value, can be controlled to within three seconds by flexibly choosing the digit capacity—for example, 21 bits for edge computing devices. The zk-SNARK proof generation, which requires multiple homomorphic addition processes associated with the complexity of the agreed polynomial, is currently completed in seconds. The proof generation of legitimate members, whose complexity depends on the chosen hash algorithm and the one-way accumulator process, can be completed in milliseconds.

Alternatively, the token verification process that occurs in the device has polynomial time complexity, and all of the above five proofs can be completed in milliseconds.

We built the experimental environment using the equipment listed in Table 2 to verify our OTMP-P2L scheme. In the PC environment, the GU100 program created 100 users, ten anonymous addresses, and key pairs for each user; the GP100 program created 100 proof strings; and the GPQR100 program created 100 QR code pictures from proof strings. In the ARM environment, the SQR100 program scanned 100 QR images using an AD703_G2J1 QR code hardware scanner, and the VP100 verified 100 proof files.

Table 3 lists the execution time of the five experimental programs; a comparison experiment was executed simultaneously on PC for VP100. In these experiments, the length of the proof string generated by GP100 is 593 B, corresponding to the size of the QR picture file, which is 23 KB. Supposing that 100 SP tokens, in the form of QR pictures, are applied every day, the security center will take up 3 MB of disk space each day.

GP100 generates a proof string that takes approximately 5 s, on average. VP100 takes approximately 0.3 s to verify a proof string in the embedded system and only approximately 0.036 s in the PC system. The number of difficult bits of workload verification can be adjusted to control the generation time of a work-load proof dynamically in the range of 1–30 s. Four random number workload calculations can also be canceled to complete verification faster for resource-constrained nodes. The scanner must accurately locate and recognize the picture to accomplish the QR code verification method; therefore, this manual alignment is slow and taking approximately 2–5 s.

We measured the CPU, memory, and IO usage of the experimental process, and collected the operating system resource usage once per second, and checked the running program. The programs take up nearly 10 MB of memory, have deficient disk IO operation, and occupy one CPU during peak computing (25% utilization). Figure 6 shows the resource usage of the three experimental processes in the PC environment. Figure 7 shows the resource usage of the two experimental processes in the ARM environment.

The experimental results show that the amount of data required for the proposed scheme, including proofs, QR code pictures, ECC keys, and other pre-made shared information, is not substantial, and they show that the resource usage of the verification calculations is within control. This system can be deployed and run stably in existing smart devices and resource-constrained IoT nodes. An open-source and common local security center software can achieve more intelligent strategies, complete multifactor authentication and multi-NP task verification, and flexibly adapt to the CIA security level requirements of different IoT systems. Moreover, the one-time token transfer and P2P blind verification mechanisms are immune to network eavesdropping and brute force. Therefore, the scheme is the best succinct authentication scheme for the IoT.

Through the verification process, we found that we should solve the following three practical problems:QR codes should be quickly scanned by light intensity, proper distance sweeping. It will be necessary to design a reasonable operating environment specification for the scheme or to study the dynamic adjustment mechanism of smartphones.Sleep and activation controls of the ARM system can minimize their power consumption. A sonar sensor, or the addition of a button for manual activation of the system, is required to detect the users’ scanning motions.The scheme should provide user-friendly prompts and verification results. For example, the system could use a speaker to output prompt sounds or integrate a digital screen for interactions.In a heterogeneous IoT environment, there are many types of devices, and more device to device communication modes need to be integrated, such as Bluetooth and sound waves. The process of automatic identification between devices is also absent. Further research should focus on possible cracking methods to find vulnerabilities.

### 6.3. Comparsion and Analysis of Schemes

Luk et al. [11] proposed seven aspects for evaluating the performance of IoT authentication mechanisms. A comparison of the OTMP-P2L scheme with existing authentication mechanisms is shown in Table 4.

The first four schemes perform authentication in a centralized cloud server, which increases network costs and makes it difficult to manage devices in a large-scale IoT. Alternatively, broadcast authentication based on a one-way accumulator performs local verification and requires only a small amount of signature data, allowing highly efficient verification. However, this authentication requires the broad-caster and the intended recipient to share a key and lacks a properly designed authentication process and a mechanism for protecting the shared key. Therefore, it risks attacks to the broadcaster and is vulnerable to a single-point failure.

On the contrary, the verification tasks in OTMP-P2L run on local nodes, slightly increasing the computing and communication overhead, however, combining multiple technologies from previous schemes to perform P2P verification, thereby preventing the risk of single-point failure. Additionally, it establishes a local security center for completing auto-registration and enhancing the authentication process, protects the synchronization of the pre-made shared secret CRS, and combines NP tasks to flexibly adjust its security level, all of which are significantly more effective measures against node hijacking and high information entropy.

## 7. Conclusions

Establishing a trust relationship is a challenging process. We envision redesigning the mechanisms of open authentication for humans and devices in the IoT from the perspective of edge computing. The focus of development should be elasticity, dynamic integration, and adjustable security, it will be more potentiality than passively accepting IoT resource-constrained to choose a lightweight authentication scheme.

The proposed OTMP-P2L scheme first constructs and saves the trusted relationship with a single program run on the smartphone or local center. The local administrator completes trusted multifactor authentication, so that subsequent one-time access tokens can be dynamically reserved at a low cost, and P2P verification can be performed succinctly. Moreover, the scheme protects privacy, manages distributed keys and shared secrets, realizes diversity, and does not have to call centralized services. The dynamic authentication with multiple correlated proofs is more secure than static multifactor identity or static tokens.

The useful study of various types of NP problems has been made to enhance the verification task. We strengthen the access control to face the cooperation and privacy issues in the IoT.

Nevertheless, more work should be carried such as accurately define each security level, choose appropriate algorithms, and determine the matching workload requirements. Currently, IoT devices rarely have high-performance ARM CPUs and there is a shortage of multimode integrated sensors. These realistic conditions will limit the evolution of intelligent verification.

Furthermore, we also should quantitatively analyze and compare various authentication schemes. If implementing a P2P network and blockchain storage, the local center will be able to maintain the consistency of the edge data. In addition, if the consistency, security and authentication algorithms can be integrated into a high-performance ASIC chip, the computational cost of this scheme will be significantly reduced. Then the local center can prevent tampering by storing and managing keys in a chip, and the chip-based intelligent recognition technology will also make the transmission of confidential data more flexible.

## Figures and Tables

**Figure 1 sensors-20-02231-f001:**
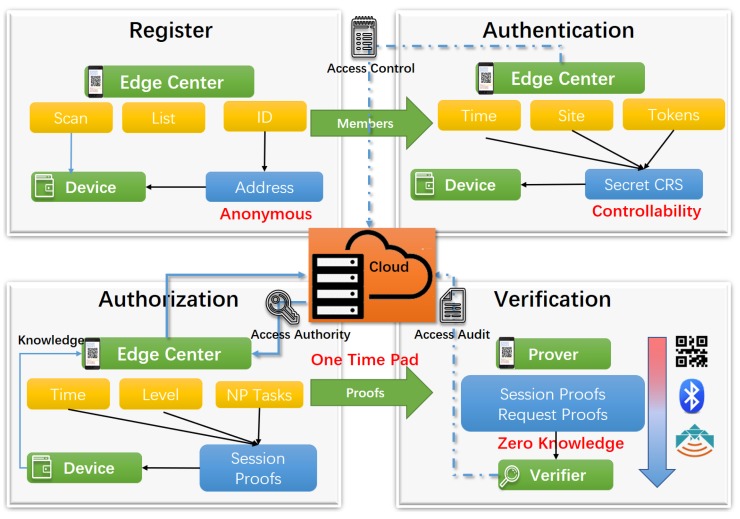
System diagram of one-time association multitask proofs for peer to local authentication (OTMP-P2L).

**Figure 2 sensors-20-02231-f002:**
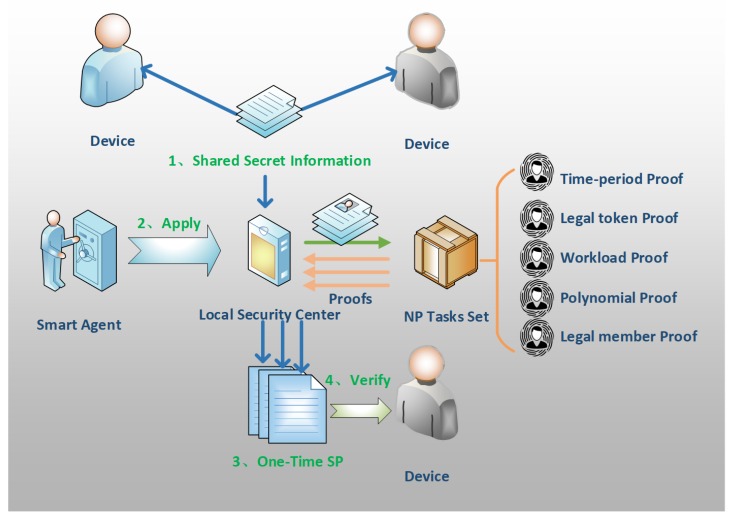
One-time association multitasking proofs for peer to local authentication (OTMP-P2L) processing procedure. Nondeterministic polynomial (NP).

**Figure 3 sensors-20-02231-f003:**
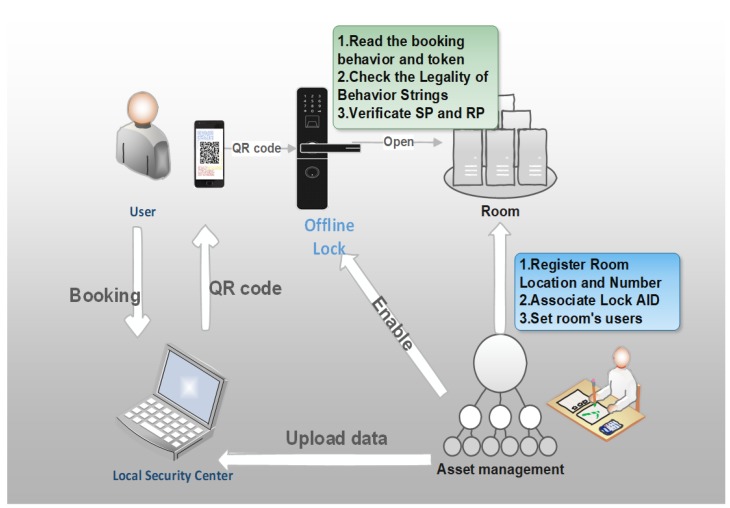
Succinct non-interactive design of the reservation unlocking process based on a QR code.

**Figure 4 sensors-20-02231-f004:**
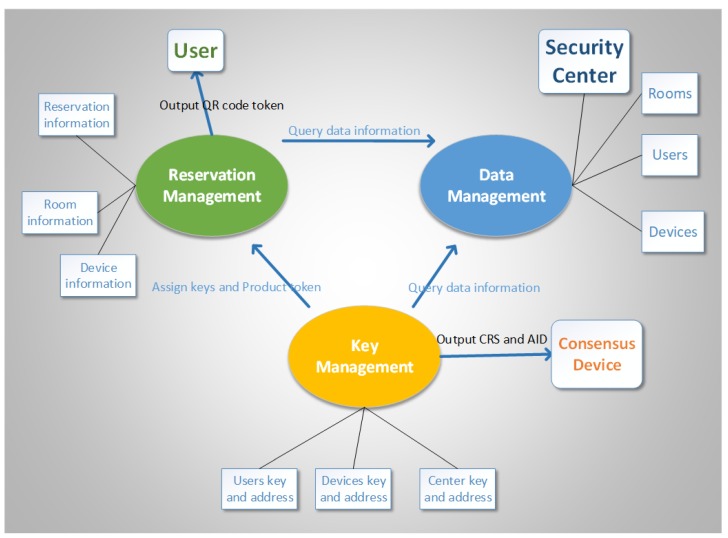
Data governance of the three-system decentralization model [31].

**Figure 5 sensors-20-02231-f005:**
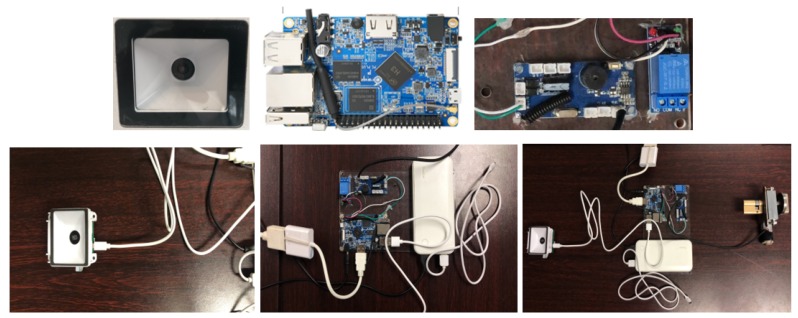
A prototype of OTMP-P2L scheme using the smart lock scenario. (**left**) AD703_G2J1 scanner. (**middle**) Main control units. (**right**) Overall circuits.

**Figure 6 sensors-20-02231-f006:**
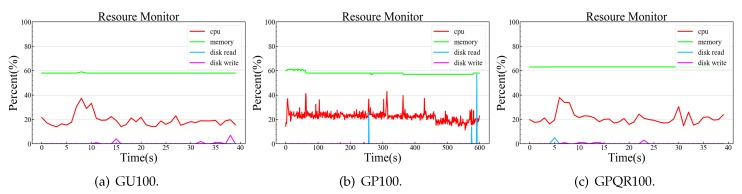
Simulating a local security center in a PC environment.

**Figure 7 sensors-20-02231-f007:**
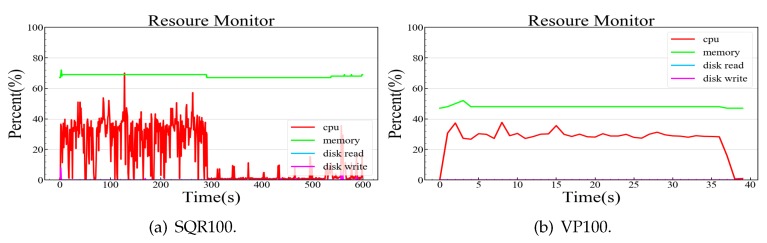
Simulating an IoT device in an advanced RISC machine (ARM) environment.

**Table 1 sensors-20-02231-t001:** Notation in the OTMP-P2L scheme.

Symbol	Notation
CRS	Common reference strings set
SP	Service proof
RP	Request proof
NPs	One-time associated multitasks set based on nondeterministic polynomial time problems
AID	Anonymous IDs associate multiple addresses and key pairs
E(x)	Homomorphic encryption algorithms
P(x)	Polynomials
zk-SNARKs	Zero-knowledge succinct non-interactive arguments of knowledge algorithms
R1CS	Rank-1 constraint system is composed of a set of mathematical form of Verification Rules like s.a∗s.b−s.c=0,(a,b,c) is a triple vector
QAP	Quadratic arithmetic programs
QSP	Quadratic span programs
CBS	Contents of the behavior string
H${}^{Nyb}$	Nyberg’s fast one-way accumulate function
A	Accumulated value
HCBS	Hash contents of the behavior string
HPC	Hash proof of work (PoW) contents
QRP	Quick response (QR) code picture

**Table 2 sensors-20-02231-t002:** Experimental device specifications.

Simulation Role	Device	Specification	Case
Security center and Provider	PC	2.6 G(MAX 3.5) Intel i7-6700HQ 4C/8T 16 G DDR3 RAM	GU100 GP100 GPQR100
Verifier	ARM	1.6 G ARM Cortex-A7 4C 1G DDR3 RAM AD703_G2J1 Scanner	SQR100 VP100

**Table 3 sensors-20-02231-t003:** The performance of the experiments.

Case	Device	Total Time (s)	Average Time (s)
GU100	PC	5.568	0.057
GP100	PC	514.027	**5.14**
GPQR100	PC	6.852	0.069
SQR100	ARM	200–500	2–5
VP100	ARM	33.851	**0.339**
VP100	PC	3.55	**0.036**

**Table 4 sensors-20-02231-t004:** System diagram of one-time multitask proofs scheme for peer to local (P2L) authentication.

Scheme	Resistance Against Node Compromise	Computation Overhead	Communication Overhead	Advantage
OAuth	High	<50 ms	>2 K	Remote center, fine-grained policy control.
DB+HMAC	Low	<50 ms	>1 K	Centralized management, the MAC address of the device is used as a unique ID.
One-Time Signature	Middle	268 ms	>1 K	Lightweight IBE-ECC algorithm, OTP is generated in the cloud.
Public-key based signature	High	>1 s	>1 K	Mature PKI system and CA center, CA unified manage certificate.
One-way accumulator	Middle	<32 ms	<132 B	Lightweight to adapt resource constraints.
Verification Phase of OTMP-P2L	High	<400 ms	593 B	One-time multitasking and local security center.

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
