# Peer review of "Trustworthiness and a Zero Leakage OTMP-P2L Scheme Based on NP Problems for Edge Security Access†"

_sensors, 2020, doi:10.3390/s20082231_

Round 1
Reviewer 1 Report
1. In Section 2, references that are not closely related to the research content can be deleted.
2. The authors don't explain well which algorithms are existed and which are innovative in Section 3 and 4.
Reviewer 2 Report
This paper presents a lightweight cryptography and authentication protocol. The proposed scheme employs one-time association multitasking
proofs for peer to local authentication namely (OTMP-P2L). In addition, The scheme chooses relevant nondeterministic polynomial problem tasks and manages localized trust and anonymity by using smart edge devices. The paper has the following problems:
- Too many buzz words: edge, authentication, cryptography. blockchain..etc. I suggest to re-write the abstract and title to become clearer.
- In related work section: "
What is Shi?Shi defined edge computin - What are the paper contributions? Add them in the introduction?
- Table of related work missing. More recent work missing too such: Blockchain for Managing Heterogeneous Internet of Things: A Perspective Architecture, Providing Secure and Reliable Communication for Next Generation Networks in Smart Cities.
- You need more discussion on the cons. algorithm used, cryptography, and testing environment.
- Discuss more on system complexity as well?
Reviewer 3 Report
In this paper, the authors proposed a new scheme that employs one-time association multitasking6 proofs for peer to local authentication (OTMP-P2L). The scheme chooses relevant nondeterministic7 polynomial (NP) problem tasks, and manages localized trust and anonymity by using smart edge8 devices such as phones and tablets, thereby enabling IoT devices to perform consensus validation9 succinctly. This consensus string is a self-consistent one-time operator that combines multiple logic10 verification rules.
The author has done a good work and this research seems to be technically sound. The research paper is good to publish as it is. Congratulations to the authors.
Author Response
Thank you very much for support. We will continue to go deep into this field.
Reviewer 4 Report
While this paper is interest, this manuscript itself requires a minor rewrite.
Some comments are listed as follows.
- The authors should present the homomorphism algorithm and the function E(x) in details. Which homomorphism algorithm was adopted in this study?
- The authors should give some case studies to explain each equation.
- The authors should provide some practical experimental results to compare the proposed method with other methods.
- In the last section, please focus on “Discussion, Implication, and Conclusion” to include
- Discussion, Implication, and Conclusion
- Discussion why the authors found out these results and how they comply (or not) with the Literature Review?
- Conclusions
- Academic Implications
- Limitations of the paper
- Future Studies and Recommendations
- The English of this paper should be polished and revised carefully, from the reviewer's point of view, the work should be written more objective and professional, things such as “we propose” should be avoided.
Round 2
Reviewer 2 Report
The authors have incorporated all my comments. I have no further comments.